# High and specific diversity of protists in the deep-sea basins dominated by diplonemids, kinetoplastids, ciliates and foraminiferans

Alexandra Schoenle [1✉], Manon Hohlfeld[1], Karoline Hermanns[1], Frédéric Mahé [2,3], Colomban de Vargas[4,5], Frank Nitsche[1] & Hartmut Arndt [1✉]

Heterotrophic protists (unicellular eukaryotes) form a major link from bacteria and algae to higher trophic levels in the sunlit ocean. Their role on the deep seafloor, however, is only fragmentarily understood, despite their potential key function for global carbon cycling. Using the approach of combined DNA metabarcoding and cultivation-based surveys of 11 deep-sea regions, we show that protist communities, mostly overlooked in current deep-sea foodweb models, are highly specific, locally diverse and have little overlap to pelagic communities. Besides traditionally considered foraminiferans, tiny protists including diplonemids, kineto-plastids and ciliates were genetically highly diverse considerably exceeding the diversity of metazoans. Deep-sea protists, including many parasitic species, represent thus one of the most diverse biodiversity compartments of the Earth system, forming an essential link to metazoans.

[1] University of Cologne, Institute of Zoology, General Ecology, Cologne, Germany. [2] CIRAD, UMR BGPI, Montpellier, France. [3] BGPI, Univ Montpellier, CIRAD, IRD, Montpellier SupAgro, Montpellier, France. [4] CNRS, Sorbonne Université, Station Biologique de Roscoff, UMR7144, ECOMAP—Ecology of Marine Plankton, Roscoff, France. [5] Research Federation for the Study of Global Ocean Systems Ecology and Evolution, FR2022/ Tara GOSEE, Paris, France. ✉email: aschoenl@uni-koeln.de; hartmut.arndt@uni-koeln.de

Although deep-sea sediment life and its extraordinary representatives have been studied for more than two centuries[1,2], we still lack a firm understanding of diversity and ecological functions in the largest ecosystem of the biosphere due to the difficulty to access it[3]. In the last two decades, the establishment of new tools for studying the molecular identity of microbial communities has revolutionized our understanding of the microbial world, and revealed a large and unique diversity of prokaryotes[4] and previously unknown protistan lineages in surface waters and the deep sea[5–7]. In parallel, morphological and molecular studies of cultured species have widened our perception of poorly represented branches of the tree of eukaryotic life[8]. Despite the fundamental roles of protists in the food web of marine surface waters[9–11], we know little about them on the deep-sea floor. Assessing deep-sea sediments' protist diversity and its biogeographic distribution is crucial to understand the ecosystem functions of eukaryotes in distinct basins, as well as the overall role of eukaryotes in global carbon cycling.

The important role of protists in energy transfer through aquatic food webs has been well established for shallow benthic and pelagic marine ecosystems[12,13], where protists have developed a wide range of nutritional strategies[14]. Within the euphotic water column, marine photosynthetic plankton forms the base of ocean food webs having a profound influence on the global carbon cycle. Protists are known as important grazers of bacteria and nutrient remineralizers in many aquatic ecosystems[9,15,16]. Delivery of fixed carbon to the deep sea via sinking detritus and carcasses provides a link between surface-associated and deep-sea detritus-based microbial food webs[17,18]. The sparse records on the functional diversity of naked and testate protists reported from the deep seafloor[7] suggests that deep-sea microbial food webs might function in a similar way as those in surface waters. Barotolerant or barophilic nanoprotists (<20 µm) may live at high hydrostatic pressure and can feed on prokaryotes in porewater as well as on those attached to particles[7]. Omnivorous protists, such as many ciliates and some rhizopods and flagellates, consume a broad spectrum of food particles including other protists and detritus. Archaeal assemblages are known to play a major role in inorganic carbon fixation in deep benthic systems[19] and at least from surface water assemblages it is known that they can form a suitable food source for protists.

Most benthic deep-sea studies has focused up to now on assumed hot spots like hydrothermal vents, cold seeps, or anoxic basins at bathyal depths ranging from 1000 to 3000 m[20–22]. There are only few studies focusing on protist communities inhabiting abyssal sediments (3000 to 6000 m depths), which cover more than half of the Earth´s surface, and even less on hadal trenches ranging from 6000 to 11,000 m depths[23–25]. Global scale comparisons, as they were made for the eukaryotic plankton community of the euphotic zone[11] or the dark ocean[26], are missing for benthic deep-sea protists.

## Results and discussion
**Deep-sea metabarcoding approach.** To explore protistan diversity in different deep-sea basins, we collected sediment samples from 20 sampling sites (3 bathyal sites, 15 abyssal sites, 2 hadal sites) in 11 regions in the Pacific and Atlantic Ocean (Fig. 1a–c, Supplementary Data 1, map created with Ocean Data View[27]). Besides sampling on a large scale to compare different deep-sea regions, we also investigated protist communities on a small spatial scale (see Supplementary Data 1). We used the approach combining DNA metabarcoding of the hypervariable V9 region of the 18S rDNA[11] and direct microscopic live observations (Fig. 1d) with cultivation of protists. Morphological and molecular characterizations of the cultures were obtained to verify

results from DNA metabarcoding, and their potential of barotolerance was also investigated[28,29]. Strict bioinformatic quality control led to a final eukaryotic dataset of ~47,000 operational taxonomic units (OTUs) (~70 million reads), of which the majority (87%) could be taxonomically assigned to groups of heterotrophic protists (Supplementary Tables 1 and 2). Keeping in mind that the number of sampled stations was more than twice as high, the eukaryotic richness in the euphotic zone of marine waters was also more than twice as high (~110,000 OTUs, the majority belonged to heterotrophic protistan groups[11]) when compared to our deep-sea eukaryotic OTUs. Within the Malaspina expedition, targeting the eukaryotic life in the deep water column, ~42,000 OTUs associated with picoeukaryotes could be recovered[30]. Protist richness in other benthic environments was lower when compared with our benthic deep-sea dataset. In the neotropical rainforests protist richness was much lower (~26,000 protist OTUs[31]). Within marine coastal sediments, the protist diversity was found to be ~6000 OTUs[32]. Comparing the number of eukaryotic deep-sea OTUs with other environments shows that the diversity of deep-sea assemblages is higher than that of coastal sediment communities and has a comparable size as the marine pelagic communities. One should keep in mind that comparing our observed protist richness with studies from other environmental biomes is difficult due to the fact that some of them used different target regions and filtering/clustering methods. Therefore, we compared the eukaryotic community of the deep seafloor with that of de Vargas et al.[11] from the sunlit ocean where similar filtering and clustering methods were used.

**Taxonomic assignment and link to deep-sea cultivable protists.** For the taxonomic assignment of sequences, we used a reference database called V9_DeepSea[33] (Zenodo, Supplementary Fig. 1 and Data 2). Besides sequences from the Protist Ribosomal Reference database PR2 v4.11.1 (ref. [34]), we included 102 in-house Sanger-sequenced strains (see Supplementary Data 2) of which the majority was isolated from deep-sea (57 strains) and marine surface waters (33 strains). We could recover 31 strains of these 102 cultivated marine protists (i.e. 21 deep-sea strains, 8 surface water strains) belonging to 20 species (19 OTUs, ~170,000 reads) with a V9 sequence similarity of 100% including Stramenopiles (bicosoecids, placidids), Discoba (kinetoplastids), Alveolata (ciliates), Obazoa (choanoflagellates), Rhizaria (cercozoans), and Cryptista (cryptophyceans). This highlights the importance of cultivation-based approaches for detailed molecular and morphological description of marine protists and the proper assignment of reads produced by NGS methods. Adding sequences from our strains increased the number of taxonomically assignable OTUs by 0.6% (273 OTUs, ~300,000 reads) with sequence similarities ranging from 80 to 100%. Overall, only 2.4% of our total protist OTUs were 100% identical to reference sequences (on average 90.4% similarity). This points to a specific and genetically distinct protist fauna in deep-sea sediments (Fig. 1d, e), which has previously been reported from studies targeting specific groups or using a smaller sampling size[20,23,25].

**High reference sequence similarity of diplonemids.** The Discoba had a higher proportion of OTUs with an overall higher similarity to reference sequences as compared to the other deep-sea protistan groups within our dataset. From the 7111 Discoba OTUs (sequence similarity ≥94%) ~89% (6300 OTUs) were associated with diplonemids. Pelagic diplonemids are depth stratified and more abundant and diverse in the deep ocean[35]. The majority of the diplonemids are thought to have a parasitic lifestyle and one possibility is that they might be not as host specific as it is known for other protists (e.g. gregarines in

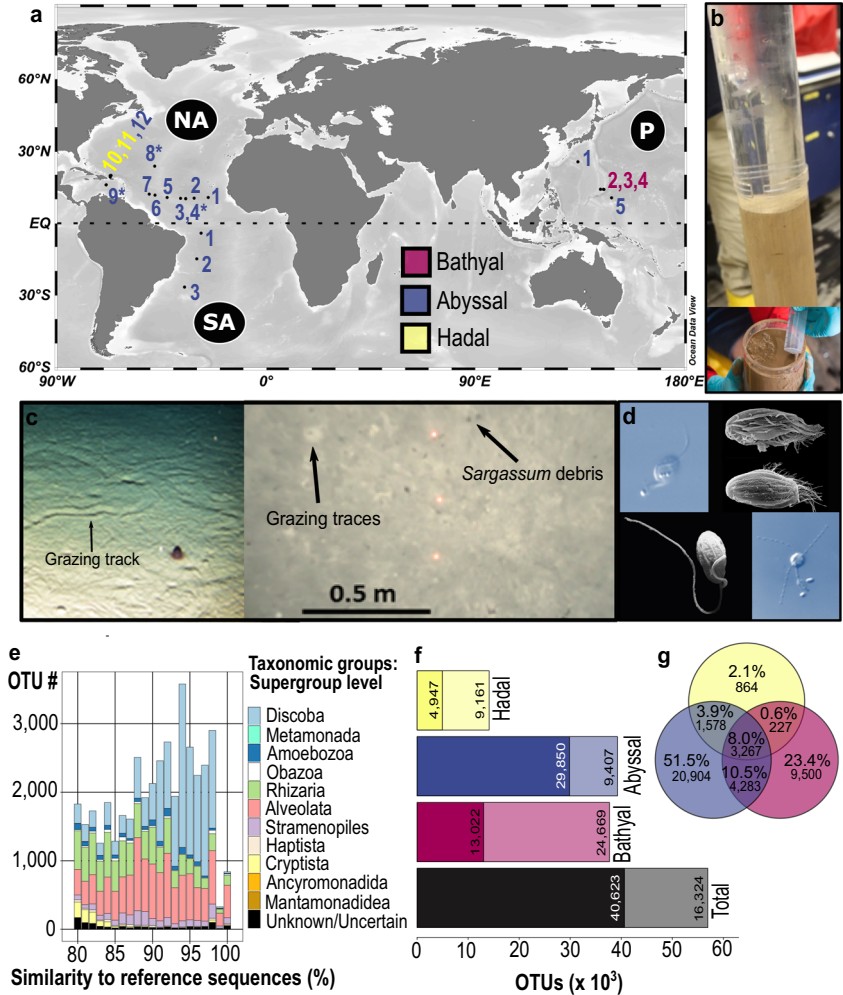

**Fig. 1 Heterotrophic deep-sea protist diversity and distribution. a** Map of the 20 examined deep-sea stations. Multiple sediment samples were analyzed separately at stations marked with an asterisk for the investigation of small-scale distribution patterns. In the North Atlantic Ocean (NA) samples were collected during two expeditions (one in summer marked with asterisk, one in winter marked with dots, map created with Ocean Data View[27]). **b** Only the upper 2 mm of undisturbed sediment from cores of the Multi-Corer were taken with a sterile syringe. **c** Images from the abyssal sea floor showing small-scale heterogeneity including grazing tracks and traces (e.g., polychaetes, holothurians), and *Sargassum* debris (© Nils Brenke, Nico Augustin, GEOMAR Kiel). **d** Observed deep-sea flagellates *Rhynchomonas nasuta* (upper left), *Euplotes dominicanus* (upper right[29]), *Keelungia nitschei* (lower left[56]), and *Massisteria marina* (lower right). **e** Similarity of heterotrophic protist rDNA richness (~22% of the total unfiltered OTUs) to total referenced eukaryotic rDNA diversity in the V9_DeepSea database. Proportion of OTUs per taxonomic group (corresponding to supergroups in the PR2 database classification) is color-coded. "Unknown/Uncertain" OTUs have been either assigned to several taxonomic division levels or to sequences taxonomically assigned only to Eukaryota. **f** Observed (dark colored) and estimated (light colored) OTUs based on the Incidence Coverage Estimator (ICE). **g** Venn diagram showing the number of unique and shared OTUs between the three different depth zones.

insects[36]). Another possibility could be that their recovery in molecular surveys might be better than for other protist lineages resulting in an over-representation in public databases. But these are only thoughts and further detailed studies of this interesting and important taxon are necessary[37,38].

**Sampling saturation and differences between depth zones.** When assessing protist diversity and saturation in our sampling effort, we could recover 71% of the total estimated sampling saturation of deep-sea heterotrophic protist OTUs by using incidence-based estimators (Fig. 1f). When considering the read abundance, saturation was nearly reached (Supplementary Fig. 2). We found great differences in OTU richness between the bathyal, the abyssal, and the hadal regions with only a small proportion of shared OTUs (Fig. 1f, g). Over half of them could only be detected in abyssal sediments, a result that might be biased by the higher sampling number of abyssal sites (Fig. 1g).

**Deep-sea eukaryotic life compared with diversity in the sunlit ocean.** A comparison with the Tara Oceans metabarcoding survey of eukaryotic diversity in the world sunlit ocean[11] revealed a fundamental difference with only a small proportion of shared OTUs with our benthic deep-sea dataset (Fig. 2 and Supplementary Fig. 3B). We found 11 hyperdiverse deep-sea protist lineages (containing ≥1000 OTUs, Fig. 2c), particularly within the Discoba (diplonemids, kinetoplastids), Rhizaria (foraminiferans), Alveolata (dinoflagellates, MALV II, MALV I, ciliates), and cryptophyceans, which accounted together for more than half of all OTUs (~56%), but only 19% of the reads. A much higher richness characterized the deep-sea diplonemid (~27.7% of the total OTUs, ~4.6% of the total reads) and kinetoplastid flagellates (~3.8% of the total OTUs, ~1.4% of the total reads), for-aminiferans (~8.2% of the total OTUs, ~2.5% of the total reads), ciliates (~6.7% of the total OTUs, ~2% of the total reads), and cryptophyceans (~2.4% of the total OTUs, ~1.8% of the total

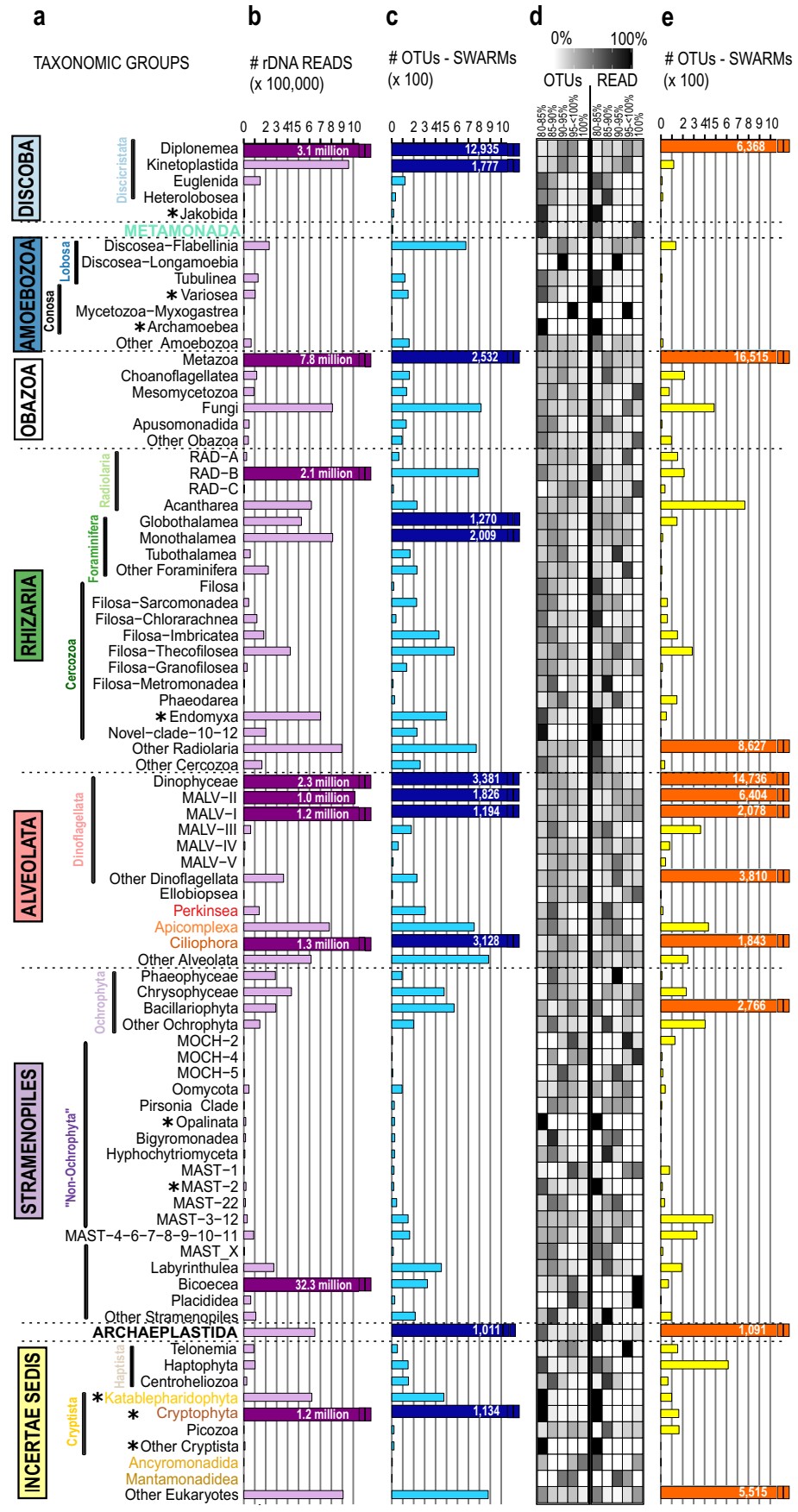

**Fig. 2 Taxonomic partitioning of the total assignable eukaryotic ribosomal diversity (V9 SSU rDNA) from the deep-sea and *Tara* Oceans[11] datasets.**
**a** Deep-branching eukaryotic taxonomic groups observed in the deep sea. Taxonomic groups include supergroups (see also Fig. 1), division (see also Fig. 3), and class/order (this figure) level as given in the PR2 database classification. Taxonomic groups, which are used in Fig. 1 (supergroups) and Fig. 3 (divisions), are colored. Asterisks indicate that >90% of reads within this lineage had a 80–85% sequence similarity to reference sequences. **b** Deep-sea eukaryotes abundance expressed as numbers of rDNA reads. Scaling of axis ranges from 0 to 1 million reads. Taxonomic groups with more than 1 million reads exceed the axis and are indicated with dark-purple bars and the number of reads is written within the bars (nine most abundant lineages with >1 million reads). **c** Deep-sea eukaryotes' richness expressed as numbers of OTUs. Scaling of axis ranges from 0 to 1000 OTUs. Taxonomic groups containing >1000 OTUs exceed the axis and are indicated with dark-blue bars and the number of OTUs is written within the bars (11 hyperdiverse lineages containing >1000 OTUs). **d** Percentage of rDNA reads and OTUs (calculated within each taxonomic group itself) with various ranges of sequence similarity (80–85%, 85–90%, 90–95%, 95–<100%, and 100%) to reference sequences. **e** Sunlit ocean eukaryotic richness expressed as number of OTUs from the *Tara* Oceans global metabarcoding dataset. Taxonomic groups containing >1000 OTUs exceed the axis and are indicated with red bars and the number of OTUs is written within the bars.

reads), as compared to their surface water relatives (Fig. 2c, e). Richness was by far the highest in diplonemids, a feature that has also been observed in deep layers of the pelagic realm[35] indicating their potential importance for deep ocean ecosystems not only in the pelagial (2.1% of the total read abundance), but also in deep-sea sediments (4.6% of the total read abundance). Local sedimentation of debris/marine snow as well as dark inorganic carbon fixation[19,39] have challenged our understanding of organic carbon available for deep-sea microbial communities[40–42]. The high number of reads associated with phototrophic species within our deep-sea dataset, e.g., within the Archaeplastida (mainly green microalgae from the family of Chloropicophyceae) and the Cryptophyta (mainly Cryptomonadales) might be due to sinking cells from surface waters down to the deep sea. On the other hand, the majority of them only had a low sequence similarity of 80–85% to Archaeplastida and Cryptophyta in the reference database and might be associated to unknown taxonomic groups especially adapted to deep-sea conditions. Several studies have reported the presence of phototrophic protists in deep waters, suggesting that mixotrophy could help them to thrive in the aphotic zone[43]. There is also the possibility for those species to enter an encysted state upon sinking[44].

***Cafeteria burkhardae* as potential global player in the marine realm**. Particularly striking was the extremely high read abundance of bicosoecids, including one OTU (~2.6 million reads) 100% identical to the species *C. burkhardae* (Fig. 2b). *C. burkhardae* was detected at all investigated deep-sea sites, matching our observation of the dominance of this species during cultivation-based approaches of deep-sea protists from several deep-sea expeditions[45]. One could argue that the occurrence of one OTU in all samples might be due to cross-sample contamination. However, sediment samples were sampled during different expeditions and the sediment was processed and analyzed separately in the laboratory. Thus, a cross-sample contamination seems to be unlikely. Interestingly, *C. burkhardae* made also a majority of the bicosoecid reads from the *Tara* Oceans surface plankton metabarcoding dataset[11,45] as well as within Malaspina metabarcoding dataset[46] targeting the water column from surface to bathypelagic waters. These occurrences in both pelagic and deep benthic ecosystems, together with recent experiments demonstrating survival at high hydrostatic pressures[47], underline the cosmopolitan distribution of selected protist species in the world's oceans across extreme environmental conditions.

**Distributional patterns of deep-sea protist richness on small and large spatial scales**. Each of the 27 sediment samples from the 11 investigated regions showed a highly distinct heterotrophic protist community (Fig. 3a) with the highest heterotrophic protist richness within the Alveolata, Discoba, and Rhizaria in each sediment sample (Fig. 3b), a pattern that has also been reported from previous bathyal and abyssal deep-sea floor studies[20,24,25]. However, diplonemids and dinoflagellates (mainly representatives of the marine alveolate (MALV) clusters) dominated the diversity at the deep seafloor (Fig. 3b). Stramenopiles (mainly bicosoecids) clearly dominated in regards of read abundances followed by high read abundances within the Alveolata, Discoba, and Rhizaria (Supplementary Fig. 4). The relative proportion of reads per sampling site and division level showed subtle differences (Supplementary Figs. 4 and 5). While the three bathyal stations from the Pacific Ocean formed a highly supported cluster, the two hadal regions from the North Atlantic Ocean clustered together with abyssal stations from the Atlantic (winter expedition) and Pacific Ocean (Fig. 3a). Furthermore, we observed distinct protist communities on much smaller spatial scale (stations NA4*, NA8*, NA9*) from sediment samples extracted just a few meters apart from each other (Fig. 3a and Supplementary Fig. 6). This could be explained by the sediment patchiness at the abyssal seafloor, which can be very high as indicated by metazoan grazing tracks, or falls of larger organic particles (e.g. debris of macrophytes, wood or dead organisms from the pelagial; Fig. 1c). The high number (~60% OTUs) of heterotrophic protists being unique to one sediment sample and the low percentage (0.6% OTUs) of heterotrophic protists shared between all samples point to the potential of highly endemic protist communities in deep-sea sediments (Fig. 3c). Such a pattern has also been found for benthic deep-sea prokaryotes in different deep-sea basins[4] and deep-sea Foraminifera[48]. The majority of "unique" heterotrophic protist OTUs had only a few reads, and several with 10–200 reads (Supplementary Fig. 7). The majority of the heterotrophic protist OTUs was represented by 16–64 reads (Supplementary Fig. 8). There was a high variation of unique protist OTUs and their taxonomic assignment per sampling site and depths (Supplementary Fig. 9). One could argue that this high dissimilarity and clustering could be the result of the high number of unique OTUs with low read abundances (Supplementary Fig. 7). However, even very conservative filtering steps (OTU abundances ≥50 or ≥100 reads) revealed a similar clustering of stations and still resulted in a great dissimilarity between protist communities on both small and large spatial scale (Supplementary Fig. 10).

**Feeding modes of deep-sea protists**. Abyssal plains are not flat or featureless, but rather strongly influenced, both by the underlying plate geology and subsequent sedimentary processes[49], which could explain that we did not observe a homogeneous deep-sea diversity pattern. The majority of taxa recorded from the different deep-sea regions belonged either to bacterivorous groups (e.g. discicristates, stramenopiles, most cercomonads, several ciliates, foraminiferans, lobose amoebae[9], or forms parasitizing other eukaryotes (e.g. perkinseans, apicomplexans, and most MALV

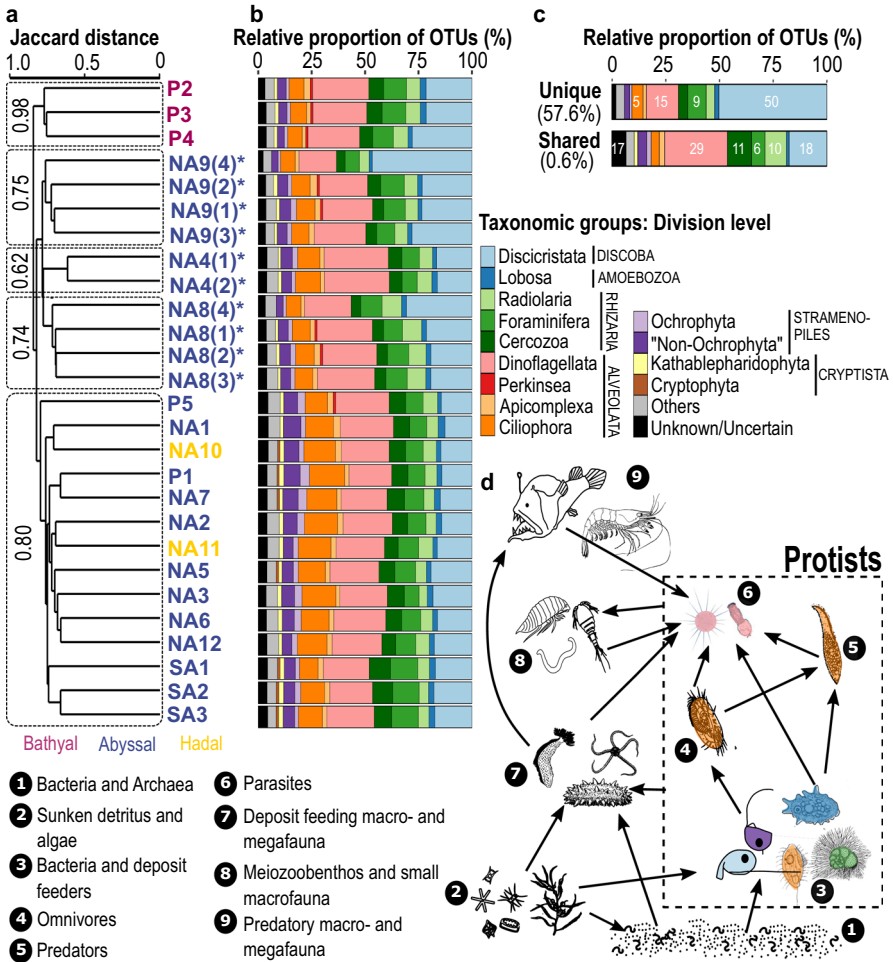

**Fig. 3 Distributional patterns and community composition of deep-sea heterotrophic protists. a** Dendrogram cluster showing the similarity (Jaccard index) of heterotrophic protist communities of the 27 sediment samples in regard to species richness based on incidence-based data (presence/absence) using UPGMA clustering. The five clusters are supported by moderate to high bootstrap values. Multiple sediment samples were analyzed separately at stations marked with an asterisk for the investigation of small-scale distribution patterns. **b** Relative proportion of OTUs within the 27 deep-sea sediment samples related to the major taxonomic protist groups. Taxonomic groups (corresponding to division level in the PR2 database classification) are only separately shown, when the number of OTUs reached more than 1% within each sample. Otherwise, OTUs were clustered together into "Others". "Unknown/Uncertain" OTUs have been either assigned to several taxonomic division levels or to sequences taxonomically assigned only to Eukaryota. **c** Relative proportion of shared (0.6%) and unique (57.6%) OTUs (heterotrophic protist richness) within all 27 sediment samples (obtained from 20 deep-sea stations). **d** Hypothetical deep-sea food web illustrating the generally ignored complex trophic interactions between microbial and macrobial components derived from their molecular diversity. Highly diverse and abundant protists are embedded in deep-sea food webs on different trophic levels as feeders on prokaryotes and particulate and dissolved organic matter, as predators, as well as parasites of metazoans and protists.

taxa among dinoflagellates). Deep-sea studies have observed protist grazing, indicating the potential of substantial reductions of the prokaryote standing stock due to protist grazing[20]. However, the quantification of protist grazing in the deep sea still needs to be investigated. Members of several groups are known to feed also on other protists (e.g. several ciliates[28,29]). Global and local differences in prokaryote diversity and abundance[4] as a main food source, endemicity of macrofauna[10,50] as important host for putative parasites might, amongst many other environmental factors varying across deep-sea habitats[50], shape deep-sea protist communities on small and large spatial scale. The impact of multiple processes and possible interactions, which might operate at the same time resulting in unique protist communities on the abyssal and hadal seafloor, still needs to be resolved.

**Role of protists in the deep-sea food web.** Our results provide a unique view on the genetic diversity and specificity of deep-sea

protist communities and point to their very important though still underestimated role in shaping seafloor communities. The estimate of heterotrophic protist species richness (Fig. 1f) for the samples from the deep-sea floor was one order of magnitude higher than that of metazoans, a tendency also obtained from the pelagial (Fig. 2 and ref. [11]). According to our data, protist communities comprise representatives of different trophic levels consisting of feeders on bacteria and archeans, on detritus, dissolved organic carbon, small eukaryotes as well as parasites of protists and metazoans (illustrated in Fig. 3d). Thus, a major part of organic carbon in deep-sea sediments is channeled not only via long known deep-sea inhabiting foraminiferans[51] but also through an unsuspected and extensive variety of small naked heterotrophic protists with different functions. These deep-sea protists form an essential link to metazoans via several trophic levels of flagellated, amoeboid, and ciliated protists by providing biochemically enriched organic matter to metazoans[40]. In

addition, due to the parasitic lifestyle of many deep-sea protists (e.g. diplonemids, MALV II[6,35]) they might act as important remineralizers of other protists and metazoans channeling carbon back to prokaryotes[8,14]. Ammonia-oxidizing Archaea have shown to dominate microbial communities in abyssal clay in the North Atlantic Ocean[52]. Due to their high abundances, Archaea should also be considered as a potential food source for deep-sea protists. In a recent study, it was shown that the probably most commonheterotrophic flagellate taxon *Cafeteria* feeds on Archaea[53]. In addition, several protists from freshwater systems have been found to positively select Archaea as food source over Eubacteria[54]. New techniques and large-scale studies, as well as long-term surveys/time series, may further elucidate the diverse composition of seafloor communities over both space and time, which is critical to our understanding of global biogeochemical cycles in the Earth's largest habitat.

## Methods

**Sampling**. The highly diverse species composition of heterotrophic protists in the deep sea demanded a combination of culture-independent (metabarcoding) and culture-dependent methods[55]. Isolation and cultivation of deep-sea protists were carried out for 102 strains to create an extended reference database (see below). In addition, eco-physiological studies were conducted for most of the strains regarding their survival at deep-sea pressure to check for their potential to belong to an active deep-sea community[28,29,45,47,56]. During four different expeditions in the Pacific and Atlantic Ocean on board of the research vessels *R/V Sonne* (SO237, SO223T) and *R/V Meteor* (M79/1, M139) sediment samples from 20 different stations (3 bathyal, 15 abyssal, 2 hadal) at 11 deep-sea basins/regions were collected using a Multi-Corer (MUC) (Supplementary Data 1). Temperature at the deep sea ranged between 2 and 4 °C; salinity was about 36 PSU. Detailed data on the conditions are available from published cruise reports of M139 (https://doi.org/10.2312/cr_m139), M79.1 (https://doi.org/10.2312/cr_m79_1), SO223T (urn:nbn:de:gbv:46-00102735-15), and SO237 (https://doi.org/10.3289/GEOMAR_REP_NS_23_2015). Subsamples of the MUC-system were taken from the upper 2 mm sediment layer by means of a sterile syringe. Only tubes with undisturbed sediment and overlaying water were used for further analyses. For 17 stations (SA1–SA3, P1–P5, NA1–NA3, NA5–NA7, NA10–NA12) taken during expeditions SO237, SO223T, and M79/1, three replicate sediment samples from three MUCs (corresponds to one core per MUC) were taken in total per station (Supplementary Data 1). For the three stations (NA4*, NA8*, NA9*) from the expedition M139, two to four replicates from three MUCs (corresponds to one to two cores per MUC) per station were taken (Supplementary Data 1). Samples were either fixated with 70% molecular biology graded ethanol and stored at −80 °C or directly deep frozen at −80 °C.

**DNA extraction, PCR amplification, and sequencing of 18S V9 rDNA metabarcodes**. Ethanol preserved sediments were treated in a speed vac for 45 min at 45 °C to evaporate the ethanol. For 17 stations (see above) taken during expeditions SO237, SO223T, and M79/1 the environmental DNA was extracted from 0.5 g sediment of each replicate sample (a total of 1.5 g per station) using the DNeasy Power Lyzer Power Soil DNA isolation kit (Qiagen, Hilden, Germany) according to the manufacturer's protocol (Supplementary Data 1). For the three stations from the expedition M139 (see above) the environmental DNA was extracted from an adapted sample volume using the same kit (Supplementary Data 1). Prior to the kit, sediment samples were pre-washed with three washing solutions to improve the success of DNA amplification by PCR in marine sediments[57]. Total DNA was quantified using a Nanodrop Spectrophotometer. For sediment samples taken during the expeditions SO237, SO223T, and M79/1, DNA of the three replicates per station were pooled in same concentrations prior to PCR amplifications. Sediment samples from the expedition M139 were separately PCR amplified without prior pooling of DNA per station to investigate small-scale patterns of deep-sea protist diversity. PCR amplifications of the hypervariable V9 region of the 18S rDNA gene was performed with the Phusion® High-Fidelity DNA Polymerase (ThermoFisher) and the forward/reverse primer-pair 1389F (5′-TTG TAC ACA CCG CCC-3′) and 1510R (5′-CCT TCY GCA GGT TCA CCT AC-3′)[58]. The PCR mixtures (25 μL final volume) contained 5 ng of total DNA template with 0.35 μM final concentration of each primer, 3% of DMSO, and 2× of GC buffer Phusion Master Mix (Finnzymes). PCR amplifications (98 °C for 30 s; 25 cycles of 10 s at 98 °C, 30 s at 57 °C, 30 s at 72 °C; and 72 °C for 10 min) of all samples were carried out with a reduced number of cycles to avoid the formation of chimeras during the plateau phase of the reaction, and in triplicates (M139) or six replicates (SO237, SO223T, and M79/1) in order to smooth the intra-sample variance while obtaining sufficient amounts of amplicons for Illumina sequencing. PCR products were checked on a 1.5% agarose gel for amplicon lengths. Amplicons were then pooled and purified using the PCR Purification Kit (Jena Bioscience, Jena, Germany). Bridge amplification and paired-end (2 × 150 bp) sequencing of the amplified

fragments were performed using an Illumina Genome Analyzers IIx system at the Cologne Center of Genomics (CCG).

**Reference database**. Due to the lack of reference sequences for the V9 region in common public databases (e.g. NCBI, PR2), we generated a dataset consisting of the V9 region of 102 marine protist strains of our Heterotrophic Flagellate Collection Cologne (HFCC), of which several have not been published yet (Supplementary Data 2). Subsamples of a few milliliters of the sediment of the MUC samples (see above) suspension were cultivated in 50 ml tissue-culture flasks (Sarstedt, Nümbrecht, Germany). Isolation was carried out using a micromanipulator or microtiter plates (liquid aliquot method[59]). All cultures were supplied with sterilized quinoa or wheat grains as an organic food source for autochthonous bacteria. After isolation, the strains were cultivated in 50 ml tissue-culture flasks (Sarstedt, Nümbrecht, Germany) filled with 30 ml Schmaltz-Pratt medium[60] (35 PSU; per liter 28.15 g NaCl, 0.67 g KCl, 5.51 g MgCl$_2$ × 6 H$_2$O, 6.92 g MgSO$_4$ × 7 H$_2$O, 1.45 g CaCl$_2$ × 2H$_2$O, 0.10 g KNO$_3$, 0.01 g K$_2$HPO$_4$ × 3H$_2$O). The cultures were stored at 10 °C in the dark. Isolates were characterized morphologically using AVEC high-resolution video microscopy and electron microscopy. For molecular studies, protistan cultures were concentrated by centrifugation (4000 × *g*, 20 min at 4 °C, Megafuge 2.0R, Heraeus Instruments). Genomic DNA of each isolated protist strain was extracted using the Quick-gDNA™ Mini Prep Kit (Zymo Research, USA). We amplified a long sequence from the 18S rDNA to the 28S rDNA with the primers 18S-For (5′-AAC CTG GTT GAT CCT GCC AGT-3′, ref. [61]) binding at the beginning of the 18S rDNA and either NLR1126/22 (5′-GCT ATC CTG AGG GAA ACT TCG G-3′, ref. [62]) or NLR2098/24 (5′-AGC CAA TCC TTW TCC CGA GTT TAC-3′, ref. [62]) binding in the 28S rDNA. PCR reactions were performed in 25 μl PCR reaction mixtures containing 5.5 μl ddH$_2$O, 1.5 units TAQ (Mastermix, VWR Germany), 2 μl DNA and 2.5 μl of each primer (forward and reverse) at a final concentration of 1.6 nM. The PCR conditions for amplifying the SSU–ITS–LSU region are as follows: pre-denaturation at 98 °C for 2 min, 35 cycles of 98 °C for 30 s, 55 °C for 45 s, and 72 °C for 4 min 30 s; final extension at 72 °C for 10 min. For bodonid strains, a different primer combination was used: 18SForBodo (5′-CTG GTT GAT TCT GCC AGT-3′, ref. [63]) + NLR1126/22 (5′-GCT ATC CTG AGG GAA ACT TCG G-3′, ref. [62]). Internal primers were used for sequencing (Supplementary Table 2). We established a new reference database for the V9 region by combining the Protist Ribosomal Reference database PR2 v4.11.1 (ref. [34]) with the 102 sequences of marine protist strains of the Heterotrophic Flagellate Collection Cologne. Using Cutadapt[64], the final in-house reference database, called V9_DeepSea[33], was trimmed to the V9 region.

**Downstream analyses and taxonomic assignment**. Our bioinformatic pipeline (adapted from Frédéric Mahé, https://github.com/frederic-mahe/swarm/wiki/Fred's-metabarcoding-pipeline) allowed filtering of high-quality V9 rDNA reads/amplicons and their clustering into OTUs (Supplementary Fig. 1). HiSeq sequencing resulted in ~223 million raw reads. Overlapping reads were assembled via VSEARCH v.2.13.4 (ref. [65]) using fastq_ mergepairs with default parameters and –fastq_allowmergestagger resulting in ~209 million assembled reads for all stations. Paired reads were retained for downstream analyses if they contained both forward and reverse primers and no ambiguously named nucleotides (Ns) using cutadapt and VSEARCH. Reads from all stations were combined in one file and de-replicated into strictly identical amplicons (metabarcodes) with VSEARCH while the information on their abundance was retained. Low abundance metabarcodes with a read abundance of one and two reads were removed from the dataset prior to OTU clustering in order to avoid potential biases associated with sequencing errors. Metabarcodes were clustered into biologically meaningful OTUs, using Swarm v2.1.5 (ref. [66]), with the parameter *d* = 1 and the fastidious option on. OTUs were taxonomically assigned to our reference database V9_DeepSea[33] using VSEARCH's global pairwise alignment and –iddef 1 (matching columns/alignment length). Amplicons were assigned to their best hit, or co-best hits in the reference database, using a pipeline called Stampa[67]. The most abundant amplicon in each OTU was searched for chimeric sequences with the chimera search module of VSEARCH, and their OTUs were removed even if they occurred in multiple samples. Sequences with a quality value (min. expected error rate/sequence length) higher than 0.0002 were discarded. Reads shorter than 87 bp were removed from the dataset. Only OTUs with a pairwise identity of ≥80% to a reference sequence were used for downstream analyses. In addition, OTUs were discarded, when a phylogenetic placement within the kingdom level was not possible. Furthermore, OTUs assigned to Metazoa, Fungi, Archaeplastida, and exclusively phototrophic organisms, including several classes of Ochrophyta (Eustigmatophyceae, Pelagophyceae, Phaeophyceae, Phaeothamniophyceae, Pinguiophyceae, Raphidophyceae, Synurophyceae, Xanthophyceae, Bacillariophyta, Chrysomerophyceae), Bacillariophytina, Filosa-Chlorarachnea within the cercozoans as well as the Cryptomonadales within the Cryptophyta, were removed (Supplementary Table 1), resulting in a final dataset of 40,623 heterotrophic protist OTUs and 55,283,811 reads. Except for Fig. 2, which compares the eukaryotic life of the deep sea with that of the euphotic zone, we used the final heterotrophic protist dataset for all graphs.

**Comparison of eukaryotic life in the sunlit ocean (Tara Ocean project) and the deep sea**. For a comparative analysis of the total eukaryotic life in the sunlit ocean to

our deep-sea NGS dataset, we downloaded the available "Database W4"[11] containing the total V9 rDNA information organized at the metabarcode (unique sequences) level from the Tara Ocean project website (http://taraoceans.sb-roscoff.fr/EukDiv/#extraction). This table contained all the 1,521,174 metabarcodes from the 47 sampled stations and the abundance information per metabarcode (in total 568,976,385 reads). We extracted this information together with the V9 sequence metabarcode and pooled these Tara Ocean metabarcodes with our deep-sea metabarcodes of 20 stations together in one file. Dereplication, clustering of metabarcodes in OTUs using Swarm, assigning the taxonomy of the representative OTU sequence to the V9-DeepSea reference database, and filtering (see steps in downstream analyses and taxonomic assignment) led to a final dataset of 123,120 eukaryotic OTUs and 589,807,407 reads. Taxonomic groups with more than 1,000 OTUs were here defined as hyperdiverse (see Fig. 2), as conducted within the framework of Tara Ocean[11].

**Statistics and reproducibility**. Stampa plots were applied to visualize our taxonomic coverage assessment to the reference database sequences. A high proportion of environmental reads assigned with a high similarity to references indicates a good coverage, while low similarity values indicate a lack of coverage[67]. Statistical analyses were conducted with R v.3.5.2 and graphs were created with the R package "ggplot2"[68]. The alpha diversity of each of the stations was assessed based on several different indices with regard to species (OTU) richness and their evenness of distribution (read abundance) including the Shannon Wiener Index, effective number of species, Simpson's Index, Pielou evenness, and Chao1 index (see Supplementary Table 2) implemented in the "fossil" package[69]. The total species richness and the species richness per depth region (bathyal, abyssal, hadal) were estimated with the incidence-based coverage estimator (ICE) using the "fossil" package. As we expected many rare species in deep-sea protist communities, we used ICE to appropriately estimate asymptotic species richness from datasets with many rare species[32,70]. Rarefaction curves were additionally used in order to investigate the degree of sample saturation by calling the function "rrafey" implemented in the "vegan" package[71]. We fit the Preston's log-normal model to abundance (read) data by calling the function "prestonfit" within the "vegan" package, which groups species frequencies into doubling octave classes and fits Preston's log-normal model. We used the function "veildedspec" to calculate the total extrapolated richness from the fitted Preston model resulting in extrapolated 44,657 OTUs. Binary-Jaccard distances were used as a measure of beta-diversity by calling the function "vegdist" within the "vegan" package. The Jaccard distance values were then used for the unweighted pair-group method with arithmetic means (UPGMA) cluster analyses ("hclust" function). Results of the cluster analyses were visualized in dendrograms by using "ggplot2". Bootstrap analyses of clusters were conducted by using the function "clusterboot" with 500,000 bootstrap replicates within the "fpc" package[72]. Venn diagrams[73,74] were used to visualize the number of shared and unique OTUs between the three depth zones and the three stations where we investigated the small-scale distribution. Heatmaps were created by using the package "pheatmap"[75]. Read abundances per division level were scaled by implementing the parameter for scaling used within the heatmap.2() package $((x - \text{mean}(x))/\text{sd}(x))$. Sample sizes and replicate details are described in the other method section parts (see also [76–82] and supplementary tables.

**Reporting summary**. Further information on research design is available in the Nature Research Reporting Summary linked to this article.

## Data availability

The data analyzed in this study are deposited at the Sequence Read Archive SRA (PRJNA635512), BioProject ID PRJNA635512, BioSamples SAMN15042370-SAMN15042370. The 18S rDNA sequences from 50 HFCC strains are deposited at GenBank under the Accession numbers MT355104–MT355153. Accession numbers of all 102 strains within our V9_DeepSea reference database[33] can be found in the Supplementary Data 2. The deep-sea reference database V9_DeepSea[33] can be downloaded from Zenodo.

## Code availability

*Data collection*: Protist Ribosomal Reference database PR2 v4.11.1 and Database W4 from the Tara Ocean project website (http://taraoceans.sb-roscoff.fr/EukDiv/#extraction). *Data analysis*: Downstream analysis of NGS raw data as described in https://github.com/frederic-mahe/swarm/wiki/Fred's-metabarcoding-pipeline and our Material and methods part. Statistical analyses were conducted with R v.3.5.2 (packages: fossil, ggplot2, vegan, fpc).

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

## Acknowledgements

We are very grateful to the Capt. Oliver Meyer, Uwe Pahl, Rainer Hammacher, and the scientific and technical crews for valuable help during sampling and the excellent support during the expeditions SO223T, SO237, M79/1, and M139. We thank Rosita Bieg, Brigitte Gräfe, and Bärbel Jendral (University of Cologne, Germany) for valuable technical support. This work was supported by grants from the Federal Ministry of Education and Research (BMBF; ProtAbyss 03G0237B and 02WRM1364) and by the German Research Foundation (DFG; AR 288/5, 10, 15, 23; MerMet 17-97; MerMet 17-11; CRC 1211 B02/03 268236062) to H.A.; C.d.V. was supported by the French Government "Investissements d'Avenir" program OCEANOMICS (ANR-11-BTBR- 0008); F.N. was supported by German Research Foundation (FN 1097/3).

## Author contributions

A.S., M.H., F.N., and H.A. were involved in the sampling of deep-sea sediment. M.H., K.H., H.A., F.N., and A.S. were involved in the cultivation and sequencing of marine protists for the reference database. M.H. and A.S. conducted the DNA extraction of sediments. A.S. conducted the bioinformatic analyses. F.M. and C.d.V. contributed data on from Tara Oceans and bioinformatics expertise. A.S. and H.A. wrote the manuscript. All authors reviewed the manuscript.

## Funding

## Competing interests

The authors declare no competing interests.
