## [Peer Review File · Communications Biology]

Reviewers' comments:

Reviewer #1 (Remarks to the Author):

comments

The authors present a study of eleven deep sea regions, using DNA meta barcoding and cultivations of protists, which shows that sediment protists are very different from water column communities with some potential site specific endemism. They used their own culture collection to update their database, which identified a lot of previously cultivated deep sea protists within the authors DNA barcoding dataset. Generally speaking it is an interesting study, based on a very high number of samples that should provide robust statistics. Probably, it is the most widely sampled study of protist diversity from deep sea sediment samples so this makes it quite novel and suitable for publication.

Here are some specific comments:

line 57 (and again on line 182): This makes it seem like the protists were already cultivated and published in two other studies (Refs 19 and 20). Please explain.

line 58: 47,000 OTUs seems extreme, if this is real then it is quite impressive. Can you discuss this in a bit more detail? How does this richness come to other protist biomes in soils or shallow sediments, or water column? This was missing in the discussion section.

line 71: That is nice, but 273 out of >40,000 OTUs is quite small. Could you please list the % of total OTUs here?

line 85: "with our" (not "to our").

line 90: What percent of the sequences and OTUs were foraminiferans? 18S primers typically miss foraminifera, so this would be nice to know.

lines 107-109: Is it possible that the detection of one OTU at all samples/sites is due to cross sample contamination in the lab? Would be good to explicitly address this with some text.

lines 169-176: It was shown recently that deep sea abyssal sediments are actually dominated by ammonia oxidizing Archaea (Vuillemin et al 2019 Science Advances), which probably are grazed by protists many of which are described in the authors study here. Thus, I imagine that many of these protists derive much of their carbon from grazing the ammonia oxidizing archaea (in addition to bacteria and other protists). Perhaps this is worth mentioning, that it is the archaea that might be the most important food source for deep sea grazing protists in abyssal sediments. Have any of the protists found in the authors study been shown to graze archaea? That would be interesting to discuss.

Reviewer #2 (Remarks to the Author):

Summary of study: Unicellular eukaryotes are overlooked in deep-sea food web ecology, but they play an important role. This study investigated the biodiversity of protists in the deep sea – specifically finding strains from deep seafloor to be distinct from what is found in reference databases, diversity in seafloor samples was patchy and species rich. Protistan species richness was especially rich relative to metazoan species richness and richness measured in pelagic samples. Protistan strains detected species included foraminifera, diplomonads, kinetoplastids, and

ciliates, many of which are known as parasites.

Overall comments:

(1) The authors have conducted an impressive study, and I think the approach to ask their questions about seafloor protistan diversity is powerful and valuable to the community. However, I would not consider the combination of reference sequences from cultures and metabarcoding to be described as a "new approach" (first appearing on l. 19, then l. 53). This approach has been used before, largely in building reference databases or adding on to a reference database so it is more appropriate for a given study system (this study). Regardless, the work is significant for the deep sea community as it represents the publication of the reference sequences of the cultured representatives.

(2) The resources to re-create the database are included, but I strongly recommend authors place the "V9_DeepSea database" into Zenodo or FigShare or via a downloadable link so it can be used by the community (this option would also have a DOI that would be cited by others and if you include an upgrade, it can link to upgrades in the database). It is a valuable database for anyone studying seafloor eukaryotes. Further, it would be helpful to also include code used in the analysis.

(3) Overall, an expansion of taxon-level observations would be helpful. Figure 2, in particular, conveys the hyperdiverse nature of some taxonomic groups and contrasts protistan community dynamics between the deep sea and Tara Ocean data. Yet, a bar plot or snapshot of composition of the deep-sea samples at sequence level, rather than OTU proportion (Fig 3b), may convey the relative importance of specific taxonomic groups contributing to the unique and shared patterns. Differences in OTU proportion in Fig 3b show subtle differences that align with the dendrogram (3a), but pairing this with sequence abundances (summed to division) would add to this story. This would also add to the thought on ll. 133-134, and ll. 135-138. Other surveys of eukaryotes in the deep-sea (hydrothermal vents) show high levels of diversity (and species richness), but these OTUs are typically found at lower abundances.

(4) Results section includes many "discussion"-like comments: some examples: l 83-85, l 95-100, l 100-114. Please modify to a combined results/discussion or move these sentences to the discussion section.

(5) Similarly, to (4), the current discussion section does not reference the importance and significance of the V9_deepsea reference dataset that the authors present. Mention of this and how the community can use it and access this would be valuable.

(6) Consider modifying the flow of the manuscript, where Figure 3 is presented before Figure 2. Figure 2 focuses on the comparison with the Tara data, while Figures 1 and 3 represent the composition of the deep sea samples with respect to the reference database and to one another (shared vs. unique).

Line by line comments:

L 34: authors list transcriptome, morphological, and molecular – I would suggest dropping transcriptome, as it is repetitive

L 40: Expansion of some of the roles eukaryotes have in carbon cycling in the introduction is recommended.

L. 61: Mention and cite PR2 here, it is the first time it is mentioned. Were additional databases used?

L. 62-63 – Sentence starting "Sanger-sequenced..." This is in reference to Data S2, please include data table citation

L. 63- What is meant by "We could recover..." I assume this is in reference to the environmental metabarcoding – meaning, authors recovered only 31 OTUs from the environmental metabarcoding data with 100% identity to the reference sequences? Please clarify and reword this sentence. Additional, L. 67 ..."and highlighting..." this should be a separate sentence, this important thought is lost as written.

L. 73-74: Should this be in reference to Fig 1e and 1f? I think this is an important finding from this work. Would authors be able to expand on Figure 1e by showing this same pattern with and without the added 103 reference strains? The values shared in the preceding paragraph are informative, but I am curious if the added 273 OTUs were dispersed across the eukaryotic tree of life, or if the added sequences 'benefited' the resolution of specific protistan lineages.

In Figure 1e, the Discoba had a higher proportion of OTUs with higher similarity to the reference database – can authors speculate on why this is?

The reference database the authors have augmented with PR2 (Data S2) has a higher number of

stramenopile sequences, did the authors observe that they recovered more stramenopile OTUs relative to other groups when using their improved deep sea database?

L. 79-82: It is unclear what is driving the unique versus shared patterns – perhaps an expansion of Figure S7 that would demonstrate the shared and unique composition of OTUs or sequence reads by site. Based on the genetic dissimilarity, the overall taxonomic composition of the OTUs and sequences should be different.

L. 83 – either in the introduction, results, or discussion, can authors please include a statement of reasoning for comparison with the Tara oceans dataset? I assume that it was to measure the level of genetic dissimilarity? The Tara oceans dataset is highly focused on the euphotic zone, favoring surface samples. A dataset like the Malaspina data would compare surface to mesopelagic protistan community composition. A short description of why the Tara oceans survey was used would be helpful.

L. 85. Define hyperdiverse threshold (in the methods section)

L. 103-104: What about the possibility of phototrophic or phototrophic-capable species to enter an encysted state upon sinking?

L. 106 – Is ribotype in reference to a single OTU?

L. 128-131: fascinating finding!

L. 134-135, “Considering all protists..” clarify what is meant by “all protist OTUs”?

L. 149-150: this study (and others) have observed protist grazing, but have not measured “substantial reductions”. Quantification of protistan grazing in the deep sea has not been published.

L. 155 – what is meant by “intensity of interactions”?

L. 158-160 and Figure 3d – depiction of a deep-sea food web with the complex interactions of protists is great.

L. 260-261 – did the use of cutadapt to modify your dataset influence the number of reference sequence included? As authors have included Sanger-sequenced references and some with other primer sets (to be expected when working with protists!). I have personally found some default tag-sequencing approach (that use cutadapt or other primer cutting/removal steps) to negatively subsample reference databases.

L. 291-293 – clarify if all figures downstream include only those ~40,000 OTUs and 55 million reads – or if some figures and analyses included all OTUs and reads.

Figures and Tables

Figure 1 – (b) insert a marker showing approximately where the 2mm cutoff is for undisturbed sediment, difficult to scale to what is shown. Note that I greatly appreciate the authors inclusion of this photo. (c) specify metazoan grazing – this was clarified in text, but inclusion here would be informative. (e) What percentage of your metabarcoding data is represented in the bar plot?

Figure 2 – (d) the % scale (top) and the placement into percentages (x-axis) was somewhat confusing. It is unclear by both percent similarity for percentages of OTUs and percentages of reads is necessary. Additionally, consider a simpler way to demonstrate this or move to supplemental.

Figure S7 – this is only showing Unique OTUs? Change plot title.

Figure S8 – describe what is meant by “Loss of OTUs %” in the figure.

Data S1 – include a column that specifies hadal, abyssal, and bathyal for each sample.

Manuscript

"Tiny naked eukaryotes dominate diversity of eukaryotes on the deep seafloor"

Alexandra Schoenle, Manon Hohlfeld, Karoline Hermanns, Frédéric Mahé, Colombran de Vargas, Frank Nitsche, Hartmut Arndt

Info: We changed the layout of the manuscript according to the journal's guidelines. Results and Discussion are now together.

REVIEWERS' COMMENTS:

Reviewer #1 (Remarks to the Author):

The authors present a study of eleven deep sea regions, using DNA meta barcoding and cultivations of protists, which shows that sediment protists are very different from water column communities with some potential site-specific endemism. They used their own culture collection to update their database, which identified a lot of previously cultivated deep sea protists within the authors DNA barcoding dataset. Generally speaking it is an interesting study, based on a very high number of samples that should provide robust statistics. Probably, it is the most widely sampled study of protist diversity from deep sea sediment samples so this makes it quite novel and suitable for publication.

Here are some specific comments:

line 57 (and again on line 182): This makes it seem like the protists were already cultivated and published in two other studies (Refs 19 and 20). Please explain.

- *Yes, some of the cultures (e.g. Cafeteria burkhardae, Pseudocohnilembus persalinus, Uronema sp. and Euplotes dominicanus) have already been published in regards of distribution and barotolerance. While they were not included in the PR2 version we were using for the metabarcoding analysis, we added them to the sequences of the PR2 version within this study and added the relevant information in the Data S2, so it is easier to follow which other sequences we included.*

line 58: 47,000 OTUs seems extreme, if this is real then it is quite impressive. Can you discuss this in a bit more detail? How does this richness come to other protist biomes in soils or shallow sediments, or water column? This was missing in the discussion section.

- *Yes, indeed, 47,000 OTUs of which ~40,000 OTUs were assigned to protists are huge. We added now information on this in the first part of the results/discussion section. However, to make a clear statement concerning the richness in other biomes is difficult due to the different target regions, sequencing methods (HiSeq, MiSeq) which give different read outputs, as well as different filtering and clustering methods. We added the following text including a reference to eukaryotic/protist richness from other marine biomes as well as from neotropical rainforest:*
- *Line 76-87: "The eukaryotic richness in the euphotic zone of marine waters was more than twice as high (~110,000 OTUs) of which most belonged to heterotrophic protistan groups¹¹. Within the Malaspina expedition, targeting the eukaryotic life in the deep water column, ~42,000 OTUs associated to picoeukaryotes could be recovered²¹. Protist richness in other benthic environments was lower when compared with our benthic deep-sea dataset. In the neotropical rainforests protist richness was much lower (~26,000 protist OTUs²²). Within marine coastal sediments, the protist diversity was ~6,000 OTUs²³. However, comparing our observed protist richness with studies from other environmental biomes is difficult due to the fact that some of them used different target regions and filtering/clustering methods. Therefore, we compared the eukaryotic community of the deep seafloor with that of de Vargas et al. from the sun lit ocean¹¹ where similar filtering and clustering methods were used."*

line 71: That is nice, but 273 out of >40,000 OTUs is quite small. Could you please list the % of total OTUs here?

- *We included the percentage of the total OTUs in the text. The sentence now reads:*
- *Line 98-99: "Adding sequences from our strains increased the number of taxonomically assignable OTUs by 0.6% (273 OTUs, ~300,000 reads) with sequence similarities ranging from 80 to 100%."*

line 85: "with our" (not "to our").

- *Thanks, we changed it accordingly.*

line 90: What percent of the sequences and OTUs were foraminiferans? 18S primers typically miss foraminifera, so this would be nice to know.

- *We added this information in the text. We also included the % of the reads and OTUs for the other groups to make it consistent. The sentence now reads:*

- *Line 114-119: “A much higher richness characterized the deep-sea diplomonad (~27.7% of the total OTUs, ~4.6% of the total reads) and kinetoplastid flagellates (~3.8% of the total OTUs, ~1.4% of the total reads), foraminiferans (~8.2% of the total OTUs, ~2.5% of the total reads), ciliates (~6.7% of the total OTUs, ~2% of the total reads) and cryptophyceans (~2.4% of the total OTUs, ~1.8% of the total reads), as compared to their surface water relatives (Fig. 2 C, E).”*

lines 107-109: Is it possible that the detection of one OTU at all samples/sites is due to cross sample contamination in the lab? Would be good to explicitly address this with some text.

- *Cross-sample contamination is always a possibility. In one of our colleagues (F.M.) experience a typical pattern is a sequence very abundant in one or a few samples (lots of reads), and present in trace amounts in other samples (a few reads). This could be due to cross-sample contamination pre-PCR, or to cross-talk between libraries during sequencing.*
- *We reconstructed our steps in the lab in this respect:*
 - *In regards of our deep sea dataset, we sampled the sediment during different expeditions. Thus, a contamination between all the stations is not possible during sampling.*
 - *We extracted the DNA of the sediment of all stations not at the same time/day/week. We splitted it in the laboratory. Thus, a contamination during this step might be neglected as well.*
 - *Moreover, PCRs were also not run in parallel for all stations. Again, we conducted PCRs on different days.*
 - *The only thing we could not influence is the step of the library preparation. We send the samples to the CCG (Cologne Center of Genomics) to sequence our samples with a HiSeq machine. They do the library prep for us (that’s obligatory).*
- *We looked at our OTU table again and had a look at the most abundant shared OTUs. The only OTU, which is highly abundant in two stations (NA1, P1, see Fig below showing the relative read abundance at each stations) is the one OTU belonging to Cafeteria burkhardae (Stramenopiles, purple colour in the figure below). We had discussed this phenomenon in our working group before. While we did almost all steps separate from one another, we think, that no cross contamination occurred during these lab steps. We once created a Mock-community from our cultured deep-sea protists, where we added to one mock community Cafeteria burkhardae and excluded it from the other mock-community. We wanted to see, if Cafeteria is highly expressed during HiSeq sequencing, although the DNA concentration was*

the same for all species. We could not observe an over-representation of Cafeteria reads in our Mock-community.

- Moreover, in other NGS datasets in our working group from 1) vertical distribution of the pelagial, 2) sediment layers (down to 3 cm)) as well as from other NGS studies e.g. Malaspina (Massana et al. 2020 ISME J) and Tara Ocean (de Vargas et al. 2015) expedition, they all found Cafeteria burkhardae to be quite abundant in their dataset. In addition, Cafeteria burkhardae appeared quite often in cultures from surface waters down to the deep sea (Schoenle et al. 2020 Europ J Protistol).
- We added the following text:
- Line 150-156: “One could argue that the occurrence of one OTU in all samples might be due to cross-sample contamination. However, sediment samples were sampled during different expeditions and the sediment was processed and analyzed separately in the laboratory. Thus, a cross-sample contamination seems to be unlikely. Interestingly, C. burkhardae made also a majority of the bicosoecid reads from the Tara Oceans surface plankton metabarcoding dataset^{11,43} as well as within Malaspina metabarcoding dataset⁴⁴ targeting the water column from surface to bathypelagic waters.”
- New figure in supplement (Fig. S4)

Fig. S4 | Community composition of deep-sea heterotrophic protists. Relative proportion of reads within the 27 deep-sea sediment samples related to the major taxonomic protist groups. Taxonomic groups (corresponding to division level in the PR2 database classification) are only separately shown, when the number of reads reached more than 1% within each sample. Otherwise, reads were

clustered together into “Others”. Unknown/Uncertain” OTUs have been either assigned to several taxonomic division levels or to sequences taxonomically assigned only to Eukaryota.

lines 169-176: It was shown recently that deep sea abyssal sediments are actually dominated by ammonia oxidizing Archaea (Vuillemin et al 2019 Science Advances), which probably are grazed by protists many of which are described in the authors study here. Thus, I imagine that many of these protists derive much of their carbon from grazing the ammonia oxidizing archaea (in addition to bacteria and other protists). Perhaps this is worth mentioning, that it is the archaea that might be the most important food source for deep sea grazing protists in abyssal sediments. Have any of the protists found in the authors study been shown to graze archaea? That would be interesting to discuss.

- *Thanks for the comment. In a recent study, De Corte et al (2019, Microb Ecol) it was shown that the probably most abundant heterotrophic flagellate taxon Cafeteria feeds on Archaea. We included this citation now. In addition, some freshwater protist species seem to prefer Archaea over Eubacteria. We discussed this possibility now and added the following text:*
- *Line 221-226: “Ammonia-oxidizing Archaea have shown to dominate microbial communities in abyssal clay in the North Atlantic Ocean⁵⁰. Due to their high abundances, Archaea should also be considered as a potential food source for deep-sea protists. In a recent study, it was shown that the probably most abundant heterotrophic flagellate taxon Cafeteria feeds on Archaea⁵¹. In addition, several protists from freshwater systems have been found to positively select Archaea as food source over Eubacteria⁵².”*

Reviewer #2 (Remarks to the Author):

Summary of study: Unicellular eukaryotes are overlooked in deep-sea food web ecology, but they play an important role. This study investigated the biodiversity of protists in the deep sea – specifically finding strains from deep seafloor to be distinct from what is found in reference databases, diversity in seafloor samples was patchy and species rich. Protistan species richness was especially rich relative to metazoan species richness and richness measured in pelagic samples. Protistan strains detected species included foraminifera, diplomonads, kinetoplastids, and ciliates, many of which are known as parasites.

Overall comments:

(1) The authors have conducted an impressive study, and I think the approach to ask their questions about seafloor protistan diversity is powerful and valuable to the community. However, I would not

consider the combination of reference sequences from cultures and metabarcoding to be described as a “new approach” (first appearing on l. 19, then l. 53). This approach has been used before, largely in building reference databases or adding on to a reference database so it is more appropriate for a given study system (this study). Regardless, the work is significant for the deep sea community as it represents the publication of the reference sequences of the cultured representatives.

- *We deleted the term “new approach”.*

(2) The resources to re-create the database are included, but I strongly recommend authors place the “V9_DeepSea database” into Zenodo or FigShare or via a downloadable link so it can be used by the community (this option would also have a DOI that would be cited by others and if you include an upgrade, it can link to upgrades in the database). It is a valuable database for anyone studying seafloor eukaryotes. Further, it would be helpful to also include code used in the analysis.

- *Yes, thanks, we will upload the V9_DeepSea database once the manuscript is gone through all publication processes. We included the DOI in the text, where we mention the reference database as well as in the section Data availability:*
- *Line 87: “For the taxonomic assignment of sequences, we used a reference database called V9_DeepSea (Zenodo doi 10.5281/zenodo.4305675, Supplement Methods, Fig. S1, Data S2).”*

(3) Overall, an expansion of taxon-level observations would be helpful. Figure 2, in particular, conveys the hyperdiverse nature of some taxonomic groups and contrasts protistan community dynamics between the deep sea and Tara Ocean data. Yet, a bar plot or snapshot of composition of the deep-sea samples at sequence level, rather than OTU proportion (Fig 3b), may convey the relative importance of specific taxonomic groups contributing to the unique and shared patterns. Differences in OTU proportion in Fig 3b show subtle differences that align with the dendrogram

- *We added a bar chart with relative abundances per station (summed per station, as done for the OTU proportion bar chart Fig. 3b) at sequence level in the supplement and added the following text in the main text:*
- *Line 166-168: “Stramenopiles (mainly bicosoecids) clearly dominated with regard to read abundances followed by high read abundances within the Alveolata, Discoba and Rhizaria (Fig. S4). The relative proportion of reads per sampling site and division level showed subtle differences (Fig. S4, S5).”*

- **Fig. S4 | Community composition of deep-sea heterotrophic protists.** Relative proportion of reads within the 27 deep-sea sediment samples related to the major taxonomic protist groups. Taxonomic groups (corresponding to division level in the PR2 database classification) are only separately shown, when the number of reads reached more than 1% within each sample. Otherwise, reads were clustered together into “Others”. Unknown/Uncertain” reads have been either assigned to several taxonomic division levels or to sequences taxonomically assigned only to Eukaryota.

(3a), but pairing this with sequence abundances (summed to division) would add to this story. This would also add to the thought on Il. 133-134, and Il. 135-138. Other surveys of eukaryotes in the deep-sea (hydrothermal vents) show high levels of diversity (and species richness), but these OTUs are typically found at lower abundances.

- *We added a bar chart and heatmap (summed to division) per station in the supplement section and added the following text in results/discussion part and materials and methods*
- *Line 166-168: “Stramenopiles (mainly bicosoecids) clearly dominated in regards of read abundances followed by high read abundances within the Alveolata, Discoba and Rhizaria (Fig. S4). The relative proportion of reads per sampling site and division level showed subtle differences (Fig. S4, S5).”*
- *Material and methods: Line 388-390: “Heatmaps were created by using the package ‘pheatmap’⁷³. Read abundances per division level were scaled by implementing the parameter for scaling used within the heatmap.2() package ((x - mean(x)) / sd(x)).”*

• **Fig. S5 | Read abundance of deep-sea heterotrophic protists. (a)** Heatmap showing the read abundance of division levels per sampling site. Reads are normalized per division level by using the following function $x - \text{mean}(x) / \text{sd}(x)$, where x is the sum of reads per division level considering all sampling sites. **(b)** Relative abundance (read number) within the 27 deep-sea sediment samples assigned to the major taxonomic protist groups on division level. Reads are set to 100% per division level considering all 27 sampling sites.

(4) Results section includes many “discussion”-like comments: some examples: | 83-85, | 95-100, | 100-114. Please modify to a combined results/discussion or move these sentences to the discussion section. (5) Similarly, to (4), the current discussion section does not reference the importance and significance of the V9_deepsea reference dataset that the authors present. Mention of this and how the community can use it and access this would be valuable.

- *We modified the text to a combined results/discussion part.*
- *We added in the material and methods part, where the reference database can be found (as you suggested we uploaded it on Zenodo).*

(6) Consider modifying the flow of the manuscript, where Figure 3 is presented before Figure 2. Figure 2 focuses on the comparison with the Tara data, while Figures 1 and 3 represent the

composition of the deep sea samples with respect to the reference database and to one another (shared vs. unique).

- *We agree that changing the order of figures would be helpful in bringing the content together. However, if you do not mind, we would like to leave it like it is now, because the last figure illustrating the food web is one of the main take home messages.*

Line by line comments:

L 34: authors list transcriptome, morphological, and molecular – I would suggest dropping transcriptome, as it is repetitive

- *We removed “transcriptome” as suggested.*

L 40: Expansion of some of the roles eukaryotes have in carbon cycling in the introduction is recommended.

- *We added the following text in the introduction:*
- *Line 40-49: “The important role of protist in energy transfer through aquatic food webs has been well established for shallow benthic and pelagic marine ecosystems^{12,13}, where protists have developed a wide-range of nutritional strategies¹⁴. Within the euphotic water column, marine photosynthetic plankton forms the base of ocean food webs having a profound influence on the global carbon cycle. Protists are known as important grazers of bacteria and nutrient remineralizers in many aquatic ecosystems^{9,15,16}. Delivery of fixed carbon to the deep sea via sinking detritus and carcasses provides a link between surface-associated and deep-sea detritus based microbial food webs^{17,18}. The sparse records on the functional diversity of naked and testate protists reported from the deep seafloor⁷ suggests that deep-sea microbial food webs might function in a similar way as those in surface waters. Barotolerant or barophilic nanoprotists (<20 µm) may live at high hydrostatic pressure and can feed on prokaryotes in porewater as well as on those attached to particles⁷. Omnivorous protists, such as many ciliates and some rhizopods and flagellates, consume a broad spectrum of food particles including other protists and detritus. Archaeal assemblages are known to play a major role in inorganic carbon fixation in deep benthic systems¹⁹ and at least from surface water assemblages it is known that they can form a suitable food source for protists.”*

L. 61: Mention and cite PR2 here, it is the first time it is mentioned. Were additional databases used?

- *Thanks, done. The text now reads: “Besides sequences from the Protist Ribosomal Reference database PR2 v4.11.1²¹, we included ...”*
- *We used only the PR2 database and 103 sequences from our own collection, which have not yet been included in this PR2 version.*

L. 62-63 – Sentence starting “Sanger-sequenced...” This is in reference to Data S2, please include data table citation

- *Thanks, done.*

L. 63- What is meant by “We could recover...” I assume this is in reference to the environmental metabarcoding – meaning, authors recovered only 31 OTUs from the environmental metabarcoding data with 100% identity to the reference sequences? Please clarify and reword this sentence.

- *Many thanks. We added the following part and hope, that it is clearer now:*
- *Line 91: “We could recover 31 strains of these 103 cultivated marine protists (i.a. 21 deep-sea strains, 8 surface water strains) belonging to 20 species (19 OTUs, ~170,000 reads) with a V9 sequence similarity of 100% including...”*

Additional, L. 67 ...”and highlighting...” this should be a separate sentence, this important thought is lost as written.

- *We separated the sentences as mentioned above.*

L. 73-74: Should this be in reference to Fig 1e and 1f? I think this is an important finding from this work. Would authors be able to expand on Figure 1e by showing this same pattern with and without the added 103 reference strains? The values shared in the preceding paragraph are informative, but I am curious if the added 273 OTUs were dispersed across the eukaryotic tree of life, or if the added sequences ‘benefited’ the resolution of specific protistan lineages.

- *While we only have to delete the 273 OTUs solely assigned to our HFCCs, and not the OTUs where references included published Accession numbers AND HFCC numbers, the output of the similarity plot was principally the same. However, the isolation and cultivation and sequencing of abundant taxa – even though contributing only a minor part to the total number of molecular diversity (OTU number) – revealed important insights from our point of view: The availability of long sequences from strains isolated from very distant places in the ocean clearly revealed that there were identical sequences of the whole 18S for at least one taxon (V9 region can sometimes not resolve to the species level). In addition, sequences were sometimes wrongly assigned in the past to species. For instance, the global distribution of the*

potentially most abundant flagellate taxon, Cafeteria burkhardae, could be verified (Schoenle et al. 2020)

- *Overall, the 273 OTUs were dispersed across the eukaryotic tree of life.*

In Figure 1e, the Discoba had a higher proportion of OTUs with higher similarity to the reference database – can authors speculate on why this is?

- *From the 7,111 Discoba OTUs with a sequence similarity $\geq 94\%$, 6,300 OTUs belonged to the Diplonemea.*
- *We added the following text:*
- *Line 103-112: “The Discoba had a higher proportion of OTUs with an overall higher similarity to reference sequences. From the 7,111 Discoba OTUs (sequence similarity $\geq 94\%$) ~89% (6,300 OTUs) were associated with diplomids. Pelagic diplomids are depth stratified and more abundant and diverse in the deep ocean ³³. The majority of the diplomids are thought to have a parasitic life style and one possibility is that they might be not as host specific as it is known for other protists (e.g. gregarines in insects ³⁴). Another possibility could be that their recovery in molecular surveys might be better than for other protist lineages resulting in an over-representation in public databases. But these are only thoughts and further detailed studies of this interesting and important taxon are necessary ^{35,36}.”*

The reference database the authors have augmented with PR2 (Data S2) has a higher number of stramenopile sequences, did the authors observe that they recovered more stramenopile OTUs relative to other groups when using their improved deep sea database?

- *Yes, the majority of the sequences in our added reference sequences belonged to Stramenopiles. Some numbers below:*
- *Within our added HFCC sequences to the reference database:*
 - *63 sequences belonged to Stramenopiles*
 - *3 sequences to Cercozoans*
 - *19 sequences to Discoba*
 - *11 sequences to Alveolata (only ciliates)*
 - *6 sequences to Obazoa*
 - *1 sequence to Cryptophyta*
- *The majority of our stramenopile sequences belonged to one species, namely Cafeteria burkhardae (bicosoecids), isolated from different depths and regions. A recent paper from our group showed, that by sequencing the type species of Cafeteria roenbergensis, almost all deposited sequences on GenBank have been falsely deposited under the name C.*

roenbergensis, because the deposited culture in CCAP was not genetically studied before. We had redescribed it to *C. burkhardae*. These stramenopile sequences in our database, thus, haven't led to a recovery of more stramenopile OTUs, because the 18S rDNA sequences were already in the reference database.

- From the 273 OTUs (~124,000 reads) **solely assigned** to our HFCCs we could recover the following groups:
 - Alveolata – Ciliophora: 39 OTUs
 - Cryptista – Cryptophyta: 8 OTUs
 - Discoba – Kinetoplastida: 87 OTUs
 - Discoba – Euglenida: 7 OTUs
 - Obazoa – Choanoflagellida: 11 OTUs
 - Obazoa – Hilomonadea: 24 OTUs
 - Rhizaria – Cercozoa: 25 OTUs
 - Stramenopiles – Bicoecia: 46 OTUs
 - Stramenopiles – Bigyromonadea: 3 OTUs
 - Stramenopiles – Chrysophyceae: 5 OTUs
 - Stramenopiles – MAST: 1 OTU
 - Stramenopiles – Placididea: 22 OTUs
 - Stramenopiles uncertain: 7 OTUs

L. 79-82: It is unclear what is driving the unique versus shared patterns – perhaps an expansion of Figure S7 that would demonstrate the shared and unique composition of OTUs or sequence reads by site. Based on the genetic dissimilarity, the overall taxonomic composition of the OTUs and sequences should be different.

- We expanded Figure S7 and added the following graph in the Supplement section:

Fig. S9 | Unique deep-sea heterotrophic protist richness amongst samples. (a) Relative proportion of unique OTUs for each of the 27 sediment samples obtained from 20 deep-sea sites (see Fig. 1). **(b)** Total abundance of unique OTUs for each of the 27 sediment samples. Numbers (%) beside bar charts represent the percentage of unique OTUs of the total number of OTUs in each sediment sample. **(c)** Relative proportion of unique OTUs within the three depth zones.

L. 83 – either in the introduction, results, or discussion, can authors please include a statement of reasoning for comparison with the Tara oceans dataset? I assume that it was to measure the level of genetic dissimilarity? The Tara oceans dataset is highly focused on the euphotic zone, favoring surface samples. A dataset like the Malaspina data would compare surface to mesopelagic protistan community composition. A short description of why the Tara oceans survey was used would be helpful.

- *We added the following text in the first part of the results and discussion section:*

- *Line 76-87: “The eukaryotic richness in the euphotic zone of marine waters was more than twice as high (~110,000 OTUs) of which most belonged to heterotrophic protistan groups¹¹. Within the Malaspina expedition, targeting the eukaryotic life in the deep water column, ~42,000 OTUs associated to picoeukaryotes could be recovered²¹. Protist richness in other benthic environments was lower when compared with our benthic deep-sea dataset. In the neotropical rainforests protist richness was much lower (~26,000 protist OTUs²²). Within marine coastal sediments, the protist diversity was ~6,000 OTUs²³. However, comparing our observed protist richness with studies from other environmental biomes is difficult due to the fact that some of them used different target regions and filtering/clustering methods. Therefore, we compared the eukaryotic community of the deep seafloor with that of de Vargas et al. from the sun lit ocean¹¹ where similar filtering and clustering methods were used.”*

L. 85. Define hyperdiverse threshold (in the methods section)

- *We added now the following sentence in the methods section:*
- *“Taxonomic groups with more than 1,000 OTUs were defined here as hyperdiverse (see Fig. 2), as conducted within the framework of Tara Ocean¹¹.”*

L. 103-104: What about the possibility of phototrophic or phototrophic-capable species to enter an encysted state upon sinking?

- *We added the following text:*
- *Line 144: “There is also the possibility for those species to enter an encysted state upon sinking⁴².”*

L. 106 – Is ribotype in reference to a single OTU?

- *We changed it to “one OTU” to make it clearer.*

L. 128-131: fascinating finding!

- *We were also surprised by this high unique fraction.*

I. 134-135, “Considering all protists.” clarify what is meant by “all protist OTUs”?

- *While we used for several graphs in the manuscript only the heterotrophic protists, we mean with “all protist OTUs” also the phototrophic protist fraction we excluded for several graphs. We hope that it is clearer now. The sentence now reads:*

- “Considering all protist OTUs (**heterotrophic and phototrophic protists**), the majority was [...]”

L. 149-150: this study (and others) have observed protist grazing, but have not measured “substantial reductions”. Quantification of protistan grazing in the deep sea has not been published.

- *Thanks, we weakened our statement. The sentence now reads:*
- *Line 196-199: “Deep-sea studies have **observed protist grazing indicating the potential of substantial reductions of the prokaryote standing stock due to protist grazing**¹². **However, the quantification of protist grazing in the deep sea still needs to be investigated.**”*

L. 155 – what is meant by “intensity of interactions”?

- *Thanks, we tried to make it more clear now:*
- *Line 192: “The **impact of multiple processes and possible interactions**, which might operate at the same time resulting in unique protist communities on the abyssal and hadal seafloor, still needs to be resolved.”*

L. 158-160 and Figure 3d – depiction of a deep-sea food web with the complex interactions of protists is great.

- *Thanks!*

L. 260-261 – did the use of cutadapt to modify your dataset influence the number of reference sequence included? As authors have included Sanger-sequenced references and some with other primer sets (to be expected when working with protists!). I have personally found some default tag-sequencing approach (that use cutadapt or other primer cutting/removal steps) to negatively subsample reference databases.

- *We agree, cutadapt can negatively subsample a reference database by discarding all sequences without primers. By default, cutadapt keeps all sequences. We used the command `–discard_untrimmed` to discard sequences without primers. Moreover, the loss of reference sequence when trimming with cutadapt can be mitigated by relaxing the max error per match, and by allowing short match overlaps (3 nucleotides by default).*

L. 291-293 – clarify if all figures downstream include only those ~40,000 OTUs and 55 million reads – or if some figures and analyses included all OTUs and reads.

- *Only Fig. 2 includes all eukaryotic groups. The rest of the graphs are only plotted with the heterotrophic protist dataset. We added the following in the methods section.*

- **Line 344:** “Except for Figure 2, which compares the eukaryotic life of the deep sea with that of the euphotic zone, we used the final heterotrophic protist dataset for all graphs.”

Figures and Tables

Figure 1 – (b) insert a marker showing approximately where the 2mm cutoff is for undisturbed sediment, difficult to scale to what is shown. Note that I greatly appreciate the authors inclusion of this photo. (c) specify metazoan grazing – this was clarified in text, but inclusion here would be informative. (e) What percentage of your metabarcoding data is represented in the bar plot?

- *(b) Yes, we understand that it is difficult to see where the 2mm cutoff is. We will definitely take a closer picture with scaling the next time we go on an expedition. For the images we use here we think, that marking this clearly in the picture is not possible due to the large size of the cores. We are really sorry, that we cannot implement your suggestion. We, thus, would like to leave the images as they are.*
- *(c) We wrote now: “(c) Images from the abyssal sea floor showing small- scale heterogeneity including grazing tracks and traces (e.g., polychaetes, holothurians, etc.), and Sargassum debris.”*
- *(e) We added now the term “heterotrophic protist rDNA richness” to make it clearer that we used the final, filtered heterotrophic protist dataset. After all filtering steps (as mentioned in the method section) we have ~22% of the total unfiltered OTUs left (see Table S1).*

Figure 2 – (d) the % scale (top) and the placement into percentages (x-axis) was somewhat confusing. It is unclear by both percent similarity for percentages of OTUs and percentages of reads is necessary. Additionally, consider a simpler way to demonstrate this or move to supplemental.

- *We would like to leave this part of the graph in the manuscript to demonstrate which groups show a higher similarity to reference sequences than others. We added now the following text in the figure description and hope, that it is clearer now:*
- **“(d) Percentage of rDNA reads and OTUs (calculated within each taxonomic group itself) with various ranges of sequence similarity (80 to 85%, 85 to 90%, 90 to 95%, 95 to <100%, and 100%) to reference sequences.”**

Figure S7 – this is only showing Unique OTUs? Change plot title.

- *Yes, thanks, we missed it. We added a graph; thus, this figure is now Fig. S9. We changed now the title to the following:*
- **“Fig. S9 | Unique deep-sea heterotrophic protist richness amongst samples.”**

Figure S8 – describe what is meant by “Loss of OTUs %” in the figure.

- *We added the following text and hope, that it is clearer now:*
- Now Fig. S10: “The column “Loss of OTUs” shows the percentage of OTUs, which did not pass the filter criterion of either ≥ 50 reads or ≥ 100 reads per OTU, considering the total OTUs (containing ≥ 3 reads) per station.”

Data S1 – include a column that specifies hadal, abyssal, and bathyal for each sample.

- *Done.*

REVIEWERS' COMMENTS:

Reviewer #1 (Remarks to the Author):

The authors have satisfactorily address all my comments, I recommend the manuscript for publication.

Reviewer #2 (Remarks to the Author):

The authors have revised the manuscript draft based on the comments from reviewers. I highly recommend that very long paragraphs be separated or include subheadings. There are many, many take home points in the combined results and discussion which are very important, but are not clearly stated. This makes the manuscript somewhat choppy.

(1) Typo in Figure 2 legend – l. 647 "(supergorups)". Note the scaling at the top x-axis, where it goes from 0-10 (x 100 or 100,000), but is written out if it exceeds 10x. I consider this a rather complicated figure, anything you can do to help the reader understand your presentation in the figure legend is recommended.

(2) The introduction section content is much improved. I would suggest separating into paragraphs – as written it is difficult to follow with many points not emphasized.

(3) The combined results and discussion are clearer than the previous version. However, subheading or sections would greatly benefit the results and discussion. For instance, the first paragraph extends from ll 65-120 – this paragraph has several take home points, but it is not organized well. It buries your core points.

(4) L. 72 change word "molecularly"

(5) Ll. 83-87 – authors go directly into a caveat instead of addressing why it is important that they recovered ~47,000 OTUs

(6) L. 105 – "higher proportion of OTUs" compared to what? Each time authors mention higher or lower (or equivalent), include what it is in comparison to. This is especially important with sequence- and OTU-derived data, as it is relative to the sequencing effort.

(7) Ll. 102-113 this could be a separate paragraph, discussing discoba findings.

(8) Suggestions on the potential ecological role that *C. burkhardae* plays?

Manuscript

"Tiny naked eukaryotes dominate diversity of eukaryotes on the deep seafloor"

Alexandra Schoenle, Manon Hohlfeld, Karoline Hermanns, Frédéric Mahé, Colombran de Vargas, Frank Nitsche, Hartmut Arndt

REVIEWERS' COMMENTS:

Reviewer #1 (Remarks to the Author):

The authors have satisfactorily address all my comments, I recommend the manuscript for publication.

- Many thanks.

Reviewer #2 (Remarks to the Author):

The authors have revised the manuscript draft based on the comments from reviewers. I highly recommend that very long paragraphs be separated or include subheadings. There are many, many take home points in the combined results and discussion which are very important, but are not clearly stated. This makes the manuscript somewhat choppy.

(1) Typo in Figure 2 legend – l. 647 "(supergorups)".

- Thanks, done.

Note the scaling at the top x-axis, where it goes from 0-10 (x 100 or 100,000), but is written out if it exceeds 10x. I consider this a rather complicated figure, anything you can do to help the reader understand your presentation in the figure legend is recommended.

- We added the following in the legend and hope it is clearer now:

Fig. 2 | Taxonomic partitioning of the total assignable eukaryotic ribosomal diversity (V9 SSU rDNA) from the deep-sea and Tara Oceans 11 datasets. (a) Deep-branching eukaryotic taxonomic groups observed in the deep sea. Taxonomic groups include supergroups (see also Fig. 1), division (see also Fig. 3) and class/order (this figure) level as given in the PR2 database classification. Taxonomic groups, which are used in Fig. 1 (supergroups) and Fig. 3 (divisions), are colored. Asterisks indicate that >90% of reads within this lineage had a 80-85% sequence similarity to reference sequences. (b) Deep-

sea eukaryotes abundance expressed as numbers of rDNA reads. Scaling of axis ranges from 0 to 1 million reads. Taxonomic groups with more than 1 million reads exceed the axis and are indicated with dark-purple bars and the number of reads is written within the bars (nine most abundant lineages with > 1 million reads). (c) Deep-sea eukaryotes' richness expressed as numbers of OTUs. Scaling of axis ranges from 0 to 1,000 OTUs. Taxonomic groups containing >1,000 OTUs exceed the axis and are indicated with dark-blue bars and the number of OTUs written within the bars (eleven hyperdiverse lineages containing >1,000 OTUs). (d) Percentage of rDNA reads and OTUs (calculated within each taxonomic group itself) with various ranges of sequence similarity (80 to 85%, 85 to 90%, 90 to 95%, 95 to <100%, and 100%) to reference sequences. (e) Sunlit ocean eukaryotic richness expressed as number of OTUs from the Tara Oceans global metabarcoding dataset. Taxonomic groups containing >1,000 OTUs exceed the axis and are indicated with red bars and the number of OTUs is written within the bars.

(2) The introduction section content is much improved. I would suggest separating into paragraphs – as written it is difficult to follow with many points not emphasized.

- We separated the text into one more paragraph.

(3) The combined results and discussion are clearer than the previous version. However, subheading or sections would greatly benefit the results and discussion. For instance, the first paragraph extends from ll 65-120 – this paragraph has several take home points, but it is not organized well. It buries your core points.

- Yes, thanks, we added subheadings for the results and discussion part (see manuscript).

(4) L. 72 change word “molecularly”

- We changed the sentence to the following „Morphological and molecular characterizations of the cultures were obtained to verify results from DNA metabarcoding...”

(5) Ll. 83-87 – authors go directly into a caveat instead of addressing why it is important that they recovered ~47,000 OTUs

- Many thanks, we included now the red labelled sentences to address this aspect.
- Keeping in mind that the number of sampled stations was more than twice as high, the eukaryotic richness in the euphotic zone of marine waters was also more than twice as high (~110,000 OTUs, the majority belonged to heterotrophic protistan groups¹¹) when compared to our deep-sea eukaryotic OTUs. Within the Malaspina expedition, targeting the eukaryotic life in the deep water column, ~42,000 OTUs

associated to picoeukaryotes could be recovered ²⁹. Protist richness in other benthic environments was lower when compared with our benthic deep-sea dataset. In the neotropical rainforests protist richness was much lower (~26,000 protist OTUs ³⁰). Within marine coastal sediments, the protist diversity was found to be ~6,000 OTUs ³¹. Comparing the number of eukaryotic deep-sea OTUs with other environments shows, that the diversity of deep-sea assemblages is higher than that of coastal sediment communities and has a comparable size as the marine pelagic communities.

(6) L. 105 – “higher proportion of OTUs” compared to what? Each time authors mention higher or lower (or equivalent), include what it is in comparison to. This is especially important with sequence- and OTU-derived data, as it is relative to the sequencing effort.

- Done. We changed the sentence to the following:
- „The Discoba had a higher proportion of OTUs with an overall higher similarity to reference sequences as compared to the other deep-sea protistan groups within our dataset.“

(7) Ll. 102-113 this could be a separate paragraph, discussing discoba findings.

- Done (see line 99, paragraph with subheading „**High reference sequence similarity of diplomemids** “), as requested above, we added several subheadings now.

(8) Suggestions on the potential ecological role that *C. burkhardae* plays?

- We separated the *C. burkhardae* part into an own paragraph with the subheading „*Cafeteria burkhardae* as potential global player in the marine realm“.